# Advanced Nanoscale Surface Characterization of CuO Nanoflowers for Significant Enhancement of Catalytic Properties

**DOI:** 10.3390/molecules26092700

**Published:** 2021-05-04

**Authors:** Muhammad Arif Khan, Nafarizal Nayan, Mohd Khairul Ahmad, Soon Chin Fhong, Muhammad Tahir, Riyaz Ahmad Mohamed Ali, Mohamed Sultan Mohamed Ali

**Affiliations:** 1Microelectronics and Nanotechnology-Shamsuddin Research Centre (MiNT-SRC), Institute for Integrated Engineering, Universiti Tun Hussein Onn Malaysia (UTHM), Parit Raja, Batu Pahat Johor 86400, Malaysia; akhairul@uthm.edu.my (M.K.A.); soon@uthm.edu.my (S.C.F.); riyaz@uthm.edu.my (R.A.M.A.); 2Department of Physics, Faculty of Basic and Applied Sciences, International Islamic University, Sector H-10, Islamabad 44000, Pakistan; shadiullahmarwat@gmail.com; 3Chemical Reaction Engineering Group (CREG), School of Chemical and Energy Engineering, Faculty of Engineering, Universiti Teknologi Malaysia (UTM), Skudai, Johor Bharu 81310, Malaysia; m.tahir@utm.my; 4School of Electrical Engineering, Faculty of Engineering, Universiti Teknologi Malaysia (UTM), Skudai, Johor Bharu 81310, Malaysia; sultan_ali@fke.utm.my

**Keywords:** nanoscale surface characterization, CuO nanoflowers, XPS, HR-TEM, SAED, HAADF-STEM, wide absorption, hydrogen peroxide, ultrathin leaves, superb catalytic performance

## Abstract

In this work, advanced nanoscale surface characterization of CuO Nanoflowers synthesized by controlled hydrothermal approach for significant enhancement of catalytic properties has been investigated. The CuO nanoflower samples were characterized by field-emission scanning electron microscopy (FE-SEM), X-ray powder diffraction (XRD), X-ray photoelectron spectroscopy (XPS), Raman spectroscopy, high-resolution transmission electron microscopy (HR-TEM), selected-area electron diffraction (SAED), high-angular annular dark field scanning transmission electron microscopy (HAADF-STEM) with elemental mapping, energy dispersive spectroscopy (STEM-EDS) and UV–Vis spectroscopy techniques. The nanoscale analysis of the surface study of monodispersed individual CuO nanoflower confirmed the fine crystalline shaped morphology composed of ultrathin leaves, monoclinic structure and purified phase. The result of HR-TEM shows that the length of one ultrathin leaf of copper oxide nanoflower is about ~650–700 nm, base is about ~300.77 ± 30 nm and the average thickness of the tip of individual ultrathin leaf of copper oxide nanoflower is about ~10 ± 2 nm. Enhanced absorption of visible light ~850 nm and larger value of band gap energy (1.68 eV) have further supported that the as-grown material (CuO nanoflowers) is an active and well-designed surface morphology at the nanoscale level. Furthermore, significant enhancement of catalytic properties of copper oxide nanoflowers in the presence of $H_{2}O_{2}$ for the degradation of methylene blue (MB) with efficiency ~96.7% after 170 min was obtained. The results showed that the superb catalytic performance of well-fabricated CuO nanoflowers can open a new way for substantial applications of dye removal from wastewater and environment fields.

## 1. Introduction

Nanostructures of copper oxide material have attracted considerable attention as a fast-developing class due to its size and shape for environmental, catalytic and energy storage applications [1,2,3,4]. Various methods have been used to determine crystal structure, shape and size of the materials, identification of elements compositions and various other physical properties of CuO nanostructures [5,6,7,8]. In many cases, there are certain physical properties that can be determined or characterized by multiple techniques. Each technique has its own strength and weaknesses, which complicate the selection of more appropriate methods. With fast development of nanotechnology, different techniques have been formulated to prepare copper oxide nanostructures with various morphologies such as nanospheres, nanoplatelets, nanopetals, nanodendrites, nanocubes, nanoribbons and nanoflowers [9,10,11,12,13]. It has been observed that only scant information exists on the formation of well-defined monodispersed flower-like morphology of CuO nanostructure materials by advanced nanocharacterization techniques [14,15,16]. The formation of CuO nanoflowers with exciting morphologies requires controlled growth parameters, because 3D hierarchical metal oxide nanostructures are still challenging due to different morphologies for various environmental applications. Also, it is essential to qualitatively and quantitatively study nanoscale material properties with detailed advanced nanoscale characterization, which would be helpful in producing specific features and extensive applications [17,18,19,20,21].

Copper oxide materials are among the best candidates due to their abundance in the environment and better electrochemical works [22,23,24,25]. It has been reported to attain superb catalytic activities in the chemical industry in many applications. The nanoscale CuO material due to its structure and well-defined morphology is treated as one of the important environmental catalysts, which are efficient to catalyze $H_{2}O_{2}$ and produce highly sensitive hydroxyl radicals (•OH) [26,27,28]. •OH is considered a better oxidizing agent and employed to decompose a large number of toxic substances without any formation of adverse products or outputs [29]. The decomposition of toxic substances with the help of copper oxide nanostructure in the presence of hydrogen peroxide is determined by surface morphology, nanocatalyst quantity, $H_{2}O_{2}$ concentration and reaction temperature of the reactants [30]. To determine well-defined surface morphology of the nanoscale structure of copper oxide material and to enhance the effect of catalysis, an accurate understanding of the nanostructure material is necessary to overcome the shortcoming for application purposes.

In the current investigation, it is noteworthy that, tailoring the dimensional morphology effects toward catalytic applications, we took advantage of 3D hierarchical copper oxide flower-like nanostructure constructed by one- or two-dimensional core components, which are fine structures, by integrating the basic core components with adding the physicochemical benefits developed by the secondary architectures. Therefore, in the present work, nanoscale surface characterization of CuO nanoflowers synthesized by controlled hydrothermal approach for significant enhancement of catalytic properties has been investigated. The structural and morphological properties of as-synthesized CuO nanoflowers were investigated by various advanced nanoscale surface characterization techniques such as XPS, HR-TEM, SAED, HAADF-STEM and STEM-EDS analysis. The phase purity of CuO flower-like nanostructures was verified by Raman and X-ray diffraction (XRD) patterns. The results of these advanced surface characterizations indicate that the hierarchical CuO nanoflowers consisted of high crystalline nature of ultrathin leaves or nanosheets. The results of UV–Vis absorption spectrums were determined to find the band gap of copper oxide nanoflowers and catalytic properties. The hierarchical CuO nanoflowers synthesized by controlled hydrothermal approach serve as active materials for significant enhancement of catalytic properties in the presence of $H_{2}O_{2}$ for the degradation of MB dye solution. The results showed that the superb catalytic performance of well-fabricated CuO nanoflowers can open a new way for substantial applications of dye removal from wastewater and environment fields.

## 2. Experimental

### 2.1. Materials

Chemicals were purchased from Sigma-Aldrich and Merck KGaA Germany and used without further purification. These chemicals include high grade copper (II) nitrate trihydrate $\left[\mathrm{Cu}\;\left({\mathrm{NO}}_{3}{)}_{2}\cdot 3H_{2}O\right)\right]$, sodium hydroxide (NaOH, 30%), hydrogen peroxide ($H_{2}O_{2}$, 30%), hexamethylenetetramine (HMTA; C_6_H_12_N_4_, ≥99.0%), ethanol (C_2_H_5_OH), and acetone (C_3_H_6_O). The water used in all synthesis procedure was ultrapure deionized water.

### 2.2. Synthesis and Growth Mechnism of CuO Nanoflowers

The hydrothermal synthesis of hierarchical copper oxide nanoflowers and their growth mechanism is shown in Scheme 1. In a typical experimental procedure, in the first step we perfectly dissolved a 5 mM quantity of copper nitrate $\left[\mathrm{Cu}\;\left({\mathrm{NO}}_{3}{)}_{2}\cdot 3H_{2}O\right)\right]$ powder in 100 mL of deionized water. In the second step, we mixed 1 mM aqueous solution of hexamethylenetetramine in 100 mL of deionized water. In the third step, the addition of solutions of steps 1 and 2 occurred and we also added 1 mL of sodium hydroxide (NaOH, 30%), as shown in Scheme 1. The final solution was kept under constant stirring for 60 min at room temperature. Consequently, after constant stirring for 1 h, the suspended mixture was transferred inside an autoclave and then properly closed. After that, the autoclave was put inside the oven for 3 h at 110 °C. After completing the whole process and cooling the oven to room temperature, the obtained solution was centrifuged and then washed with deionized water. At the end, a black colour (CuO nanoflowers) product is obtained after drying in a vacuum oven at 60 °C for 2 h. Scheme 1 clearly illustrates the complete growth of CuO nanoflowers through hydrothermal method.

From the experimental results presented in Scheme 1, the growth mechanism of CuO nanoflowers can predict the following. It should be noted that, during the course of mixing of copper nitrate and HMTA, there was no direct precipitation, but the light-blue colored solution of copper nitrate changed into turbid by the addition of HMTA; furthermore, when we added 1 mL of sodium hydroxide (NaOH, 30%), as shown in Scheme 1. The final solution was kept under constant stirring for 60 min time at room temperature. Interestingly, it was observed that the solution was immediately changed from turbid to blue-colored due to an immediate formation of $\mathrm{Cu}\;{\left(\mathrm{OH}\right)}_{2}$ nuclei. Sodium hydroxide immediately enables to obtain and produce OH^−^ ions for the development of $\mathrm{Cu}\;{\left(\mathrm{OH}\right)}_{2}$ nuclei according to the chemical reaction below.
(1)$$\mathrm{Cu}\;{\left({\mathrm{NO}}_{3}\right)}_{2}\cdot 3H_{2}O+2\mathrm{NaOH}\to \mathrm{Cu}\;{\left(\mathrm{OH}\right)}_{2}+2{\mathrm{NaNO}}_{3}+3H_{2}O$$

The formation of $\mathrm{Cu}\;{\left(\mathrm{OH}\right)}_{2}$ is very significant for the growth of CuO crystallites, which initially assist as building blocks for the constitution of the final products (CuO crystallites). At proper heating and time duration, the aggression of $\mathrm{Cu}\;{\left(\mathrm{OH}\right)}_{2}$ nuclei starts to become CuO crystallites according to the chemical reaction given below.
(2)$$\mathrm{Cu}\;{\left(\mathrm{OH}\right)}_{2}\;\Delta \to \;\mathrm{CuO}+H_{2}O\;$$

With the individual CuO nanopetals starting at the initial heating stage, but with the longer reaction times up to 3 h, the initially formed CuO nanopetals were assembled and form flower-like morphologies. We also observed NaOH acting as a strong electrolyte for neutralizing the surface charge of CuO and may be helpful in aggregating the individual petals for the formation of flower-like morphology. The electrostatic attractions due to strong binding and good concentration of HMTA performs an important role in the aggregation of individual CuO nanopetals to form well-defined monodispersed flower-like morphologies. Also, it was reported that, during reaction, the function HMTA can be understood in two ways: first, it can be hydrolyzed in the distilled water, and secondly, it slowly produced OH^−^ ions, as shown by the chemical reaction given below.
(3)$$C_{6}H_{12}N_{4}+6H_{2}O\to 6HCHO+4{\mathrm{NH}}_{3}$$
(4)$${\mathrm{NH}}_{3}+H_{2}O\to {\mathrm{NH}}^{+}{}_{4}+{\;\mathrm{OH}}^{-}$$

Therefore, it is concluded that continuous supply of Cu^2+^ ions and OH^−^ ions from the copper precursor and HMTA, respectively, leads to the continuous generation of $\mathrm{Cu}\;{\left(\mathrm{OH}\right)}_{2}$ units, which finally converts into CuO crystallites and forms the flower-shaped CuO nanostructures.

### 2.3. Characterization

Morphological observations and crystallinity of the as-grown CuO nanoflowers are characterized by field emission scanning electron microscopy attached with EDS, Geol diffractometer (XRD), Raman Spectroscopy and high-resolution transmission electron microscopy (JEOL JEM-ARM 200F). The chemical composition of the surface/interface of CuO flower-like nanostructure, including its purity and oxidation state, are determined by X-ray photoelectron spectroscopy (Shimadzu Kratos Axis Ultra DLD, XPS) using current 10 mA at 15 kV. The optical and catalytic properties of the as-synthesized CuO nanoflowers is determined by UV–Vis spectrometers (Schimadzu 1800).

### 2.4. Catalytic Measurement

The catalytic action of as-synthesized CuO nanoflowers in the oxidation of MB was measured in the presence of $H_{2}O_{2}$. For the measurement of catalytic properties of CuO nanoflowers, firstly we measured the stability of MB dye solution in the presence of hydrogen peroxide only, and then the oxidation of MB in copper oxide nanoflowers was measured in the presence of $H_{2}O_{2}$. In the experimental procedure of catalytic measurements, 20 mg of the as-synthesized copper oxide nanoflowers in powder form was mixed with 100 mL of an aqueous solution of methylene blue (0.2 g/L). Additionally, 20 mL of hydrogen peroxide $\left(H_{2}O_{2}\right)$ solution was added to the mixture. The final solution $\left(\mathrm{MB}+H_{2}O_{2}+\mathrm{CuO}\right)$ is kept under constant stirring for 60 min at room temperature to react properly and to reach equilibrium of the final product solution. Then, at regular time intervals, a 3.5 mL quantity of the final solution was taken in quartz cuvette and examined with the help of UV–visible spectrophotometer. The peak intensity for the standard curve for MB solution concentrations was obtained at ~λ=664.25 nm.

## 3. Results and Discussion

### 3.1. Surface Morphology

Figure 1 shows FE-SEM images and EDX analysis of CuO nanoflowers synthesized by the controlled hydrothermal approach. The low magnified FE-SEM images of copper oxide nanoflowers are presented in Figure 1a,b, which clearly shows that the substrates’ surfaces were covered by a large quantity of homogenous and uniform-sized flower-like nanostructures. The 3D hierarchical copper oxide flower-like self-assembled nanostructures with well-defined morphology have been recognized. The high-magnification FE-SEM image of the single CuO nanoflower structure is shown in Figure 1c, which reveals that the flowers consist of several ultrathin leaves. The diameters of the ultrathin leaves are different from the roots to the tips. The CuO flowers have large roots with sharpened tips. The large roots of the ultrathin leaves are connected mutually in one center in such a way to make attractive morphology of 3D hierarchical flower-like structure. The monodispersed individual CuO nanoflower diameter is about ~4–5 µm. Furthermore, the ultrathin leaves or petals of CuO nanostructures provide a large surface-to-volume ratio for mass transport property of catalytic applications. Additionally, perfect size and shape of copper oxide nanoflowers were examined with high-resolution TEM measurement (vide infra). Figure 1d shows the EDX result and the compositional study of CuO nanoflowers, which clearly indicate that CuO nanoflowers consist of only Cu and O elements.

### 3.2. Crystallinity

The crystallinity of copper oxide nanoflowers was examined by X-ray diffraction technique. The X-ray diffraction patterns of the as-synthesized samples of CuO flower-like nanostructures are shown in Figure 2. It has been clearly shown that all diffraction peaks were well indexed to the pure monoclinic CuO flower-like nanostructures. No impurity peaks were observed. All peaks in the diffraction pattern were matched with the standard CuO phase of a monoclinic structure of reference code ICSD 98-004-3179. The first strongest peak was noticed at 2θ value of 35.6°, which is related to the planes (002) and (11-1), while the second strongest peak was noticed at 2θ value of 38.8°, which is related to the planes (111) and (200). The appearance of the two strongest peaks according to the reference code further supports the pure phase of monoclinic CuO nanoflowers. The confirmation achieved from X-ray diffraction pattern shows that our preparation process is perfect without any impurity, controllable and can reduce defects in comparison to a closed system. The crystallite size and average crystallites size of CuO nanoflowers calculated from the XRD data using Scherrer Equation are given below.
(5)D=Kλ/βcosθ
where,
D = Crystallite size (nm)K = 0.94 (Scherrer constant)λ = 0.15406 nm (Wavelength of the X-ray source)β = FWHM (radians)θ = Bragg’s angle in degrees

The broadening of all the XRD peaks in the spectrum shows the presence of nanoscale crystallites. The calculated average crystallites size of CuO nanoflowers from XRD data using Scherrer Equation is 24.35 nm, which shows the results in nanoscale of high surface-to-volume ratio.

### 3.3. X-rays Photoelectron Spectroscopy (XPS) Study

The XPS study is a more effective surface characterization technique for the confirmation of nanostructures material composition, oxidation state and purity of the material [31,32,33,34]. The XPS measurement used a monochromatic source with Al Kα radiation having photon energy of 1486.6 eV, which was operated at 10 mA and 15 kV for the surface analysis of CuO flower-like nanostructure and their binding energy spectrums. The C 1s peak at 284.60 eV was utilized as a standard reference for the measurements of all the binding energies. Figure 3a shows that the Cu 2p high-resolution energy scans on the sample presented two main peaks having binding energies (BE) at 933.4 and 953.5 eV, which are respectively related to $\mathrm{Cu}\;2p_{3/2}$ and $\mathrm{Cu}\;2p_{1/2}$ of CuO nanoflowers [35,36]. The splitting between $\mathrm{Cu}\;2p_{3/2}$ and $\mathrm{Cu}\;2p_{1/2}$ states is ~20 eV, which confirms the formation of CuO phase. The $\mathrm{Cu}\;2p_{3/2}$ at 933.4 eV has a satellite peak with a doublet at ~942 and ~944 eV, which is in good agreement with the provided information in the literature [37]. Figure 3b presents the XPS spectrum of oxygen (O 1s) element having three peaks located at 529.12 eV, 530.76 eV and 532.18 eV, which are related to the lattice oxygen ${\left(O_{L}\right)}^{2-}$, oxygen vacancies ${\left(O_{V}\right)}^{2-}$ and absorbed oxygen on the surface of CuO nanoflowers, respectively. The observed XPS profile clearly shows the absence of ${\mathrm{Cu}}_{2}O$, and also $\mathrm{Cu}{\left(\mathrm{OH}\right)}_{2}$ impurities within the investigated sample, which confirms the existence of CuO.

### 3.4. Raman Analysis

Figure 3c shows the room temperature Raman spectrum of copper oxide nanoflowers. CuO belongs to the $C_{2h}^{6}$ space group and has a monoclinic structure. There are three acoustic modes $\left(A_{u}+2B_{u}\right)$, six infrared active modes $\left(3A_{u}+3B_{u}\right)$, and three Raman actives modes $\left(A_{g}+2B_{g}\right)$. These Raman modes are shown in Equation (6) [38,39,40,41].
(6)$$\Gamma_{RA}=4A_{u}+5B_{u}+A_{g}+2B_{g}$$

It is considered that copper oxide nanoflowers are composed of three Raman active optical phonons at 297, 346 and 632 cm^−1^, which are shown in Figure 3c. We can attribute the high intensity peak at 297 cm^−1^ to the $A_{g}$ mode and the other two peaks at 346 and 632 cm^−1^ to the $B_{g}$ modes. The Raman analysis peaks position of copper oxide nanoflowers prepared by controlled hydrothermal method at low temperature has confirmed the phase purity with a monoclinic structure and no other phase or impurities are revealed.

### 3.5. TEM Analysis

Higher-resolution transmission electron microscopy (HR-TEM) is one of the most advanced surface characterization techniques to study the perfect morphology and crystal quality of as-synthesized individual CuO nanoflower in greater depth. Figure 4a–f shows the TEM and HR-TEM images from low magnification to high magnification of the as-synthesized individual/single CuO flower-like nanostructure, which are in good agreement with the obtained FE-SEM results. The average diameter of the flower-like structure was in the range of 4–6 µm, as shown in Figure 4a. Figure 4b–d has shown clearly that individual flower-like nanostructure consists of several ultrathin nanoleaves or nanopetals. The diameter of the ultrathin leaves differs from the roots to the tips. The CuO flower has the large roots with sharpened tips. The large roots of the ultrathin leaves are connected mutually in one center in such a way as to make an attractive morphology of a 3D hierarchical flower-like structure. Figure 4d shows that the length of one ultrathin leaf of copper oxide nanoflower is about ~650–700 nm, while the base is about ~300.77 ± 30 nm. Figure 4d,e has determined that that the average thickness of the tip of an individual ultrathin leaf of copper oxide nanoflower is about ~10 ± 2 nm. Figure 4e–g shows HR-TEM images, which indicates that the ultrathin nanoleaves had good crystallinity along with the lattice spacing of 0.27 nm, which relates to the (110) lattice fringe of monoclinic CuO, as corroborated by XRD analysis. This result was also supported by the X-ray diffraction pattern. From the selected area electron diffraction (SAED) pattern as shown in Figure 4h, discrete spots with electron diffraction rings reveal that copper oxide flower-like nanostructures are crystalline in nature and possess the monoclinic phase of CuO. Figure 4i shows high-angle annular dark field (HAADF) scanning TEM analysis of single CuO nanoflower for atomic-resolution characteristic image. Both the high-resolution TEM and SAED analysis support the Raman and XRD results. The results of TEM, HR-TEM and HAADF-STEM have clearly revealed that the shape and size of the as-grown material (CuO nanoflowers) is a good crystalline and well-designed surface morphology at the nanoscale level.

### 3.6. Advanced STEM Analysis (HAADF-STEM)

The important surface characterization tool of high resolution TEM analysis is used to yield atomic-resolution characteristic images, which gives crystal information at atomic scale at surface/interface known as bright field image or high-angular annular dark field (HAADF)-STEM mode [42,43]. The elemental composition is provided by STEM-EDS of single CuO nanoflower. Figure 5a shows HAADF scanning transmission electron microscope (STEM) image of a single copper oxide nanoflower. The elemental composition of CuO nanoflowers for copper and oxygen elements are shown in Figure 5b,c. Figure 5d shows the STEM energy dispersive spectrums (STEM-EDS) or elemental mapping composition of individual CuO nanoflower. The elemental mapping composition of individual copper oxide nanoflower supported the existence of copper (Cu) and oxygen (O) elements with similar CuO nanoflower distribution. The result of STEM analysis has confirmed the purity and elemental mapping of individual element of CuO nanoflower.

### 3.7. Optical and Catalytic Properties

The optical and catalytic properties of copper oxide nanoflowers were examined by using a UV–visible absorption spectroscopy technique for absorption spectrum, band gap determination and time profile degradation of methylene blue in the presence of hydrogen peroxide spectrums. The hierarchical CuO nanoflowers synthesized by controlled hydrothermal approach serve as active materials for significant enhancement of catalytic properties in the presence of $H_{2}O_{2}$ for the degradation of MB dye solution. The UV–visible absorption of CuO nanoflowers is shown in Figure 6a. Enhanced absorption of visible light ~850 nm of CuO flower-like nanostructures morphology provided a better support for catalytic application. Figure 6b shows the band gap energy of CuO nanoflowers determined from Tauc plot. The band gap energy of copper oxide nanoflowers was obtained to be ~1.67 eV, which clearly indicates larger values than bulk copper oxide (1.24 eV) material. The larger value of band gap energy (1.68 eV) has further supported that the as-grown material (CuO nanoflowers) is an active and well-designed surface morphology at the nanoscale level.

Copper oxide material nanostructures have been reported to attain superb catalytic activities in the chemical industry in many applications. In the present study, common cationic dye methylene blue (MB) was selected as organic contaminant or pollutant. For the catalytic test measurements of CuO nanoflowers, firstly we measured the stability of MB dye solution in the presence of $H_{2}O_{2}$ only and it was determined that MB dye degradation was only ~7%, which shows greater stability in the dark. Figure 6c indicates absorption spectrums of methylene blue in the presence of CuO nanoflowers and $H_{2}O_{2}$ at different reaction times. It has been clearly shown that methylene blue absorption peak at 664.25 nm decreased with reaction time, as shown in Figure 6c. In the final analysis, the absorption peak became very wide and weak at reaction time 170 min, and no other visible absorption peak was recognized, indicating the nearly complete or final stage of MB degradation. Figure 6d,e presents the concentration ratio and percentage concentration ratio plots in the presence of $H_{2}O_{2}$ versus reaction time. The percentage degradation formula is given in Equation (7).
(7)$$ƞ=\left(1-\frac{S_{t}}{S_{0}}\right)\times 100\%$$
where ƞ = % degradation, $S_{t}$ = absorption at a given reaction time and $S_{0}$ = absorption at t=0.

Furthermore, the kinetics of the methylene blue degradation in the presence of hydrogen peroxide and copper oxide nanoflowers is shown in Figure 6f. The apparent rate constant (k) of this degradation was determined from the slope of ln(S_t_/S_0_) versus reaction time, which is found to be ~0.0196 min^−1^.

The maximum percentage decomposition of MB dye in the presence of $H_{2}O_{2}$ and copper oxide nanoflowers was found to be ~96.7% with the reaction time being 170 min (Figure 6e). The significant enhancement of catalytic properties of copper oxide nanoflowers in the presence of $H_{2}O_{2}$ for the degradation of MB dye solution is compared in Table 1 with other research reported in the literature. The results showed the superb catalytic performance of well-fabricated CuO nanoflowers due to the supportive performance of $H_{2}O_{2}$ and advantage of 3D hierarchical copper oxide flower-like nanostructures. Furthermore, 3D hierarchical showed copper oxide flower-like nanostructures constructed by one- or two-dimensional core components, which are fine structures by integrating the basic core components with adding the physicochemical benefits developed by the secondary architectures.

The proposed possible mechanisms of CuO nanoflowers to facilitate the degradation of methylene blue in the presence of hydrogen peroxide without any external source light are given in the following two steps.

First step:$$\left[\mathrm{Cu}\left(\mathrm{II}\right)\right]+H_{2}O_{2}⇌\left[\mathrm{Cu}\left(\mathrm{II}\right)\right]\ldots .H_{2}O_{2}\to \left[\mathrm{Cu}\left(I\right)\right]+•\mathrm{OOH}+H^{+}$$
$$\left[\mathrm{Cu}\left(I\right)\right]+H_{2}O_{2}⇌\left[\mathrm{Cu}\left(I\right)\right]\ldots .H_{2}O_{2}\to \left[\mathrm{Cu}\left(\mathrm{II}\right)\right]+•\mathrm{OH}+{\mathrm{HO}}^{-}$$

Second step:$$•\mathrm{OOH}+H_{2}O_{2}\to H_{2}O+•\mathrm{OH}+O_{2}$$
$$•\mathrm{OH}+H_{2}O_{2}\to H_{2}O+•\mathrm{OOH}$$

After adsorbing MB molecule and $H_{2}O_{2}$ on the surface of the copper oxide nanoflowers, $H_{2}O_{2}\;$reacts with the complex surface of the flowers [Cu(II)], which produce as a result free radical •OOH and species [Cu(I)]. Further reaction with$\;H_{2}O_{2}$ oxidized back to [Cu(II)] in conjunction with radical •OH as shown in the first step. The second step shows that the free radicals may once again be adsorbed on $H_{2}O_{2}$ and produce free radicals of radical •OH, •OOH or •O^2−^. The free radicals produced as a result of the first step and second step are responsible for very high oxidizing ability to interact with MB dye, and therefore greatly enhance the oxidative degradation rate of methylene blue dye. Therefore $H_{2}O_{2}$ can be decomposed by CuO into free radical species •OH, •OOH or •O^2−^, which have a much stronger oxidation power than $H_{2}O_{2}$, and therefore remarkably improve the oxidative degradation of MB

In addition, the catalytic test for the degradation of methylene blue in the presence of hydrogen peroxide and CuO NFs catalyst were run for three times only and the degradation rate (%) was slightly changed from 96.7% to 95%, which shows no significant change for the degradation of methylene blue. This evidence provided the confirmation that CuO NFs have high stability.

## 4. Conclusions

In conclusion, advanced nanoscale surface characterization of CuO Nanoflowers synthesized by controlled hydrothermal approach for significant enhancement of catalytic properties has been performed. The structural and morphological properties of as-synthesized CuO nanoflowers were investigated by various advanced nanoscale surface characterization techniques such as XPS, HR-TEM, SAED, HAADF-STEM, STEM-EDS and Raman analysis. The result of HR-TEM has shown that the length of one ultrathin leaf of copper oxide nanoflower is about ~650–700 nm, base is about ~300.77 ± 30 nm and the average thickness of the tip of an individual ultrathin leaf of copper oxide nanoflower is about ~10 ± 2 nm. The results of this advanced surface characterization have shown that the hierarchical CuO nanoflowers consisted of highly nanocrystalline ultrathin leaves with monoclinic structure. 3D hierarchical copper oxide flower-like nanostructure constructed by one- or two-dimensional core components, which are fine structures by integrating the basic core components with adding the physicochemical benefits developed by the secondary architectures, have been investigated. The phase purity of CuO nanoflowers has been investigated by STEM-EDS, Raman and XRD analysis. Enhanced absorption of visible light ~850 nm and larger value of band gap energy (1.68 eV) have further supported that the as-grown material (CuO nanoflowers) is an active and well-designed surface morphology at the nanoscale level. The significant enhancement of catalytic properties of copper oxide nanoflowers in the presence of $H_{2}O_{2}$ for the degradation of MB dye solution with efficiency ~96.7% after 170 min was obtained, which is greater than those reported in the literature. The results showed that the superb catalytic performance of well-fabricated CuO nanoflowers can open a new way for substantial applications of dye removal from wastewater and environment fields.

## Data Availability

Not applicable to this article.

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
