# Peer review of "Advanced Nanoscale Surface Characterization of CuO Nanoflowers for Significant Enhancement of Catalytic Properties"

_molecules, 2021, doi:10.3390/molecules26092700_

Round 1
Reviewer 1 Report
The manuscript proposed by Nafarizal Nayan and co-workers, entitled “Advanced Nanoscale Surface Characterization of CuO Nanoflowers for Significant Enhancement of Catalytic Properties” reports on the development of the fine morphology of CuO nanoflowers that were hydrothermally synthesized, characterized by various techniques and catalytically evaluated with regard to the degradation of methylene blue in the presence of hydrogen peroxide. It is of great significance that the authors examine a cheap transition metal oxide, such as copper oxide. Generally, the manuscript is well-structured and it is worth being published in Molecules. However, there are some major points that need to be addressed before publication.

Author Response
First of all, we would like to thank the editorial board and all the respectable reviewers of the “Molecules” journal for their valuable comments and suggestions. In light of these useful comments, we have tried to improve our manuscript. We have made corrections and revised the manuscript. The detailed responds to reviewer comments is attached.

Reviewer 2 Report
The paper “molecules-1167218- Advanced Nanoscale Surface Characterization of CuO Nanoflowers for Significant Enhancement of Catalytic Properties” investigated CuO Nanoflowers synthesized by controlled hydrothermal approach for significant enhancement of catalytic properties. Research on nanoflower containing CuO has been very limited, and studies on nanoparticles seem to be valuable. Nanoflowers preparation, SEM, XRD, XPS and TEM tests report is acceptable. The article layout and figure descriptions are OK. The manuscript is interesting; however, to improve the quality, the following recommendations can be incorporated.
* The logic of the introduction is not clear enough so that the reviewer is confused to some degree. In order to descript the process and significance of present object clearly, please clarify the technical logic of present object to improve the introduction.
* The language of the paper needs to be improved, as such it is really difficult to read...
* The abstract lacks key findings and contribution of the study to the body of knowledge. Too much background information in the abstract. Put only one or two lines to present the background of the problem and then present the method, findings and contribution of the study.
* The quality of figures 1, 5 and 6c are poor and needs to describe the target of this figure in this manuscript.
* The introduction should be rewritten to show the highlights and novelty of the work.
* The results are superficial. The authors should try to provide more explanations.
We recommend publication after revision. We congratulate the authors on a substantial contribution to the scientific discourse.
Author Response

(The authors gave the same response as above.)

Round 2
Reviewer 1 Report
The authors have considerably enhanced the quality of the revised manuscript and therefore, it should be published in the present form.
Reviewer 2 Report
Title: Advanced Nanoscale Surface Characterization of CuO Nanoflowers for Significant Enhancement of Catalytic Properties
In this revised manuscript, the Authors have made corrections according to referee comments. In my opinion, the manuscript in current form could be considered for acceptance.